# Rice β-Glucosidase 4 (Os1βGlu4) Regulates the Hull Pigmentation via Accumulation of Salicylic Acid

**DOI:** 10.3390/ijms231810646

**Published:** 2022-09-13

**Authors:** Peizhou Xu, Tingkai Wu, Asif Ali, Jinhao Wang, Yongqiong Fang, Runrun Qiang, Yutong Liu, Yunfeng Tian, Su Liu, Hongyu Zhang, Yongxiang Liao, Xiaoqiong Chen, Farwa Shoaib, Changhui Sun, Zhengjun Xu, Duo Xia, Hao Zhou, Xianjun Wu

**Affiliations:** 1Key Laboratory of Southwest Crop Genetic Resources and Genetic Improvement, Ministry of Education, Rice Research Institute, Sichuan Agricultural University, Chengdu 611130, China; 2Rubber Research Institute, Chinese Academy of Tropical Agricultural Science, Haikou 571101, China; 3Key Laboratory of Ministry of Education for Genetics, Breeding and Multiple Utilization of Crops, College of Agriculture, Fujian Agriculture and Forestry University, Fuzhou 350002, China; 4Department of Plant Breeding and Genetics, University of Agriculture, Faisalabad 38000, Pakistan

**Keywords:** salicylic acid, SA *O*-β-glucoside, flavonoids, anthocyanins, hull pigmentation, photosynthetic rate

## Abstract

Salicylic acid (SA) is a stress hormone synthesized in phenylalanine ammonia-lyase (PAL) and the branching acid pathway. SA has two interconvertible forms in plants: SAG (SA *O*-β-glucoside) and SA (free form). The molecular mechanism of conversion of SA to SAG had been reported previously. However, which genes regulate SAG to SA remained unknown. Here, we report a cytoplasmic β-glucosidase (β-Glu) which participates in the SA pathway and is involved in the brown hull pigmentation in rice grain. In the current study, an EMS-generated mutant *brown hull 1 (bh1)* displayed decreased contents of SA in hulls, a lower photosynthesis rate, and high-temperature sensitivity compared to the wild type (WT). A plaque-like phenotype (brown pigmentation) was present on the hulls of *bh1*, which causes a significant decrease in the seed setting rate. Genetic analysis revealed a mutation in *LOC_Os01g67220*, which encodes a cytoplasmic Os1βGlu4. The knock-out lines displayed the phenotype of brown pigmentation on hulls and decreased seed setting rate comparable with *bh1*. Overexpression and complementation lines of *Os1βGlu4* restored the phenotype of hulls and normal seed setting rate comparable with WT. Subcellular localization revealed that the protein of *Os1βGlu4* was localized in the cytoplasm. In contrast to WT, *bh1* could not hydrolyze SAG into SA in vivo. Together, our results revealed the novel role of *Os1βGlu4* in the accumulation of flavonoids in hulls by regulating the level of free SA in the cellular pool.

## 1. Introduction

Salicylic acid (SA), also known as o-hydroxybenzoic acid, acts as a signaling molecule for plant disease response [1]. SA widely exists in animals, plants and microorganisms and is required for several key processes of systemic acquired resistance [2]. SA plays an important role in protecting normal growth and development and helps the plant to cope with extreme environments. Its biosynthesis mainly depends on two pathways: the first is the branching acid pathway, which is the main SA synthesis pathway in plants and animals. In this pathway, SA is produced by an iso-branching acid synthase [3]. The second is the phenylalanine ammonia-lyase (PAL) pathway, in which cinnamic acid is catalyzed to produce SA after forming a benzoic acid [4]. The PAL pathway also forms other important derivatives, such as flavonoids and lignin [5]. The active form of SA is limited in plants, as most of SA is modified to its derivatives e.g., SAG and SA glucose-ester (SGE) after glycosylation, methylation, hydroxylation, methyl salicylate and others, etc. [6,7]. When plants need SA to defend against external disturbance, SAG and Me-SA will rapidly decompose to produce an active (free) form of SA [3].

Flavonoids have attained more attention because of their importance in biological functions. Anthocyanins absorb visible light to promote photosynthesis and are also considered essential for the coloration of flowers and fruits [8,9]. Flavonoids also play a role in the interaction between animals and plants. For example, the astringency of leaves is due to the presence of polyamines, which prevents herbivores from nibbling on plants [10]. Phenolic compounds have also been reported to show variation during ripening in the skin of *Vitis vinifera* L. [11]. Some studies have shown that pollen fertility and development are regulated by the transport of content of flavonoids and auxin [12,13,14,15]. A rice β-glucosidase, *Os12βGlu38*, has been reported to play its role in the pollen fertility and development of the intine cell wall [16]. In addition to their important role in controlling plant physiological development, flavonoids also play a protective role against cardiovascular diseases [17]. Flavanol and anthocyanins accumulate in the leaf epidermis and bind to DNA to form complexes that protect plants from oxidative damage [18,19,20]. Similarly, anthocyanins in maize and Arabidopsis seedlings were significantly accumulated under low-temperature conditions, indicating their potential role in low-temperature stress [20]. The anabolism of phenolic substances is important for plant development and also plays a crucial role in the SA pathway, which is not studied well in monocots.

A ternary complex of R2R3-MYB, basic helix-loop-helix (bHLH) and WD40 transcription factors (TFs) have been reported to regulate the structural enzymes of the flavonoid biosynthesis pathway [21,22]. Several genes encoding these TFs, especially bHLH and WD40, have already been identified. MYB TF is an N-terminal MYB domain containing protein and consists of more than 50 amino acids (R1, R2, and R3), and its functions are mainly determined with the specificity of R repeats. The MYB TF e.g., C1 (COLORLESS 1) and PL1 (PURPLE LEAF 1) are reported to regulate the flavonoid pathway [23]. The bHLH TF, also known as MYC, is a regulator of the flavonoid pathway. These are named after due to the presence of a bHLH conserved domain and are widely distributed in plants, yeast and humans [21,24]. WD40 or WDR (WD repeat) TFs are transcriptional complexes and are involved in many eukaryotic cellular processes, including cell division, vesicle formation and transportation, signal transduction, RNA processing, and transcriptional regulation [25]. WD TFs are particularly involved in the remodeling of chromatin and histones and indirectly affect transcription events. The WD40 proteins are not thought to have any catalytic activity (DNA binding or regulating the expression of target genes) and appear to be a docking platform because they can interact with several proteins simultaneously [26]. Collectively, the ternary complex of R2R3-MYB, bHLH and WD40 regulate the multiple enzymatic steps of the flavonoid pathway. 

Rice hull is a protective covering of rice grain and can be brown, red, black, and purple. However, modern cultivars have no hull pigmentation, and their hulls are of a white-straw color [27]. Various genes have been reported to regulate the coloration of floral organs by anthocyanin and flavonoid pathways [28]. A recent study has indicated that the C-S-A gene system regulates the hull pigmentation in rice by the mutual cascade of the *C1* (R2R3 MYB), *S1* (bHLH) and *A1* gene [27]. In this module *C1*, *S1* and *A1* work together for the coloration of the hull as a color-producing gene, tissue-specific color controlling gene, and an encoder of flavonoids, respectively. 

β-d-glucoside glycohydrolase, also known as β-glucosidase (β-Glu), releases terminal glucosyl residues from glycosides and oligosaccharides [29]. Based on amino acid sequence and structural similarity, glucoside hydrolases were divided into 133 families [30]. Plant glucosides regulate the physiological activity of phytohormones, which are stored in their inactive glycosylated forms [31]. In plants, β-Glu is involved in various biological and developmental processes. One of the important functions of β-Glu is the induction of deglycosidase to regulate the level of active hormones [32]. β-Glu hydrolyzes a glucoside to release an active form of phytohormone, although only a few enzymes have been reported to play such functions [33]. Glycoside hydrolase function varies due to substrate specificity and tissue localization [34]. A rice, β-Glu, Os4BGlu13 hydrolyzes 1-O-β-D-glucosyl ester, tuberonic acid and SA-derivatives of glycosides [35]. Osβ4Glu12, Os3βGlu6 and Os4βGlu12 hydrolyze different substrates and play a role in defense response, the ABA pathway, photosynthesis and the degradation of cell wall-derived oligosaccharides [36,37,38]. A previous study has demonstrated that *Os1βGlu4* plays a role in synthesizing 4-nitrobenzene-oligosaccharides and alkyl-glycosides in vitro [39]. However, the specific physiological function and mode of action of *Os1βGlu4* have yet to be explored in plants. 

We identified an EMS-generated mutant *brown hull1 (bh1)*, which shows the brown pigmentation (plaque-like phenotype) on the hull’s surface of grains. The characterization and functional analysis of *Os1βGlu4* were supported by physiological, morphological and molecular analysis. Here, we presented the novel role of *Os1βGlu4* in the pigmentation of the hull by regulating the level of free SA in the cellular environment. 

## 2. Results

### 2.1. bh1 (Brown Hull1) Displayed Brown Hull Pigmentation and Decreased Seed Setting Rate

We identified a *bh1* mutant from the EMS mutagenized library. The phenotypic observations revealed that *bh1* started to show the visible accumulation of brown pigmentation on hulls after one week of the heading stage. At the same time, wild-type (WT, Yixiang1B) did not accumulate any pigmentation. Initially, the hulls of *bh1* developed a light brown color, turning into dark brown as the plant gradually moved to the maturity stage (Figure 1a–c). The caryopsis of *bh1* did not show the accumulation of any pigmentation; however, the whole panicle of *bh1* gives a brownish appearance, while that of the WT panicle gives a normal white straw-colored appearance (Figure 1d–h). Under normal growth conditions, grain length and width of the *bh1* were smaller than that of WT by 13.17% and 12.45%, respectively (Appendix A). Other major agronomic yield traits, such as plant height (13.45%), 1000-seed weight (−18.60%), seed setting rate and grain number per panicle (69.84%), were significantly decreased compared to WT (Appendix A).

### 2.2. Hulls of bh1 Have Decreased Activity of Photosynthesis and Sugars

A previous study revealed that the chlorophyll (Chl) contents of leaves and glume directly affect a plant’s photosynthesis and agronomic yield [40]. To determine whether the accumulation of pigmentation has affected the photosynthetic activity in *bh1*, we quantified Chl-a and Chl-b pigments at the start of the booting stage in the hulls of WT and *bh1* (Figure 2). These results showed *bh1* did not show significant changes compared to WT (Figure 2a). We compared the Chl contents in *bh1* before and after the visible appearance of brown pigmentation and compared them to WT at the same stage (Figure 2b,c). Chl-a and Chl-b pigments were significantly lower in *bh1* after the appearance of pigmentation. However, anthocyanins were significantly increased in the hulls of *bh1* after the pigmentation compared to the WT (Figure 2c). The net photosynthetic rate (NPR) of *bh1* was 38.59% decreased compared to WT (Figure 2e). At the same time, soluble sugars and saccharose contents were significantly decreased in *bh1* (Figure 2f,g). Together, these results suggested that decreased NPR, total soluble sugars, and saccharose contents after the pigmentation were the reason for the low seed setting rate of *bh1.*

### 2.3. Accumulation of Reactive Oxygen Species (ROS) in bh1 Panicles

The accumulation of glume plaques (brown pigmentation) is often accompanied by ROS, leading to programmed cell death (PCD) in plants. To detect whether the hulls of *bh1* have accumulated ROS, malondialdehyde (MDA), and hydrogen peroxide (H_2_O_2_) contents were quantified before and after the pigmentation (Figure 3a,b). The contents of MDA and H_2_O_2_ were significantly increased in *bh1* after the pigmentation compared to before the pigmentation. The activities of CAT and SOD enzymes were also significantly increased after the pigmentation in the hulls of *bh1* (Figure 3c,d). The results showed MDA, H_2_O_2_, CAT, and SOD MDA were not significantly changed compared to WT before the pigmentation. However, the contents of MDA and H_2_O_2_ were increased three- and twofold, respectively, in *bh1* after the appearance of brown pigmentation. A TUNEL assay also revealed the PCD and DNA fragmentation due to the increased activity of ROS in the hulls of *bh1* (Figure 3e). The glume of *bh1* showed more positive fluorescent signals (green) of TUNEL compared to WT. Together, these results revealed that *bh1* hulls pigmentation was associated with the increased ROS and other CAT and SOD enzymes.

### 2.4. Genetic Analysis Reveals a Cytoplasmic Os1βGlu4 Encodes bh1 Phenotype

For genetic analysis of *bh1,* two mapping populations were constructed by crossing *bh1* with WT (*indica*) and 02428 (*japonica*). In both populations, all F_1_ plants showed normal white-straw hulls, however, some F_2_ plants showed brown pigmentation on hulls with a segregation ratio of 3:1 (χ^2^ = 0.64 < χ^2^_0_._05_ = 3.57). The segregation ratios showed that a single recessive gene controls the phenotypic trait of pigmentation in *bh1*. 456 SSR markers were individually applied to the pooled DNA of F_2_ populations for primary gene mapping. The gene controlling *bh1* phenotype was initially located in the long arm of chromosome 1 between markers Os1-20 and Os1-21 (Figure 4a). The gene was further narrowed down to the region between RM7650 and RM3632. Due to the further unavailability of more indels, a Mutmap assay was conducted to find mutations according to the method discussed in a previous study [41]. Consistently, the Mutmap assay revealed an SNP in the region, which was previously identified through SSR markers. According to the Mutmap assay, a nucleotide (671st) in the fifth exon of *LOC_Os01g67220* was changed from C to T, resulting in the change of amino acid (234th) from glycine to valine (Figure 4b,d). This candidate gene was amplified from both WT and *bh1*, and the results were consistent with the Mutmap. Based on the data mentioned above, *LOC_Os01g67220* was selected as the candidate gene for the mutant phenotype of *bh1*.

According to the Rice Genome annotation Project and annotation analysis, *LOC_Os01g67220* encodes an *Os1βGlu4,* and its protein length is composed of 483 amino acids. It has 97.1% sequence similarity with the domain of β-Glu and Glycoside hydrolase based on sequence similarity. A total of 35 active β-glucosidases were found in the rice genome database, and the evolutionary tree was constructed based on the amino acid similarity (Appendix A) These 35 family members were divided into eight clusters (Os1-8) according to the genetic distance. The Os1βGlu4 was categorized into the fourth cluster (Os4), and other closely related family members were Os3βGlu6, Os10βGlu34, and Os6βGlu25. A study has already reported that *Os3βGlu6* had strong hydrolytic activity for PNP-β-d-glucoside [37]. At the same time, chloroplastic Os3βGlu6 also contributes to cellular ABA pools and affects drought tolerance and photosynthesis in Rice [38]. In addition, previous studies have purified *Os1βGlu4* protein in vitro and conducted protein experiments in vitro. Similarly, cytoplasmic Os1βGlu4 can preferentially hydrolyze PNP-β-d-glucoside and oligosaccharides in vitro [39]. Sequence alignment and phylogenetic analysis reveal that the candidate gene of *bh1* encodes an Os1βGlu4, and previous studies indicate that it has a strong hydrolyzing activity for β-d-glucoside. Hence, we tentatively named the candidate gene of the *bh1* phenotype as “*Os1βGlu4*”.

The transgenic assays of complementation and knock-out verified the pigmentation phenotype of Os1βGlu4. 

To verify the biological function of *Os1βGlu4* in rice, a knock-out target site was designed in the first 25 bp of the candidate gene (*LOC_Os01g67220*) through clustered regularly interspaced short palindromic repeats (CRISPR/Cas9). Genetic transformations were carried out in both the japonica (Nipponbare) and indica (WT, Yixiang) cultivars (Figure 5a,b). The positive transgenic lines (Ko^Nip^ and Ko^WT^) revealed the pigmentation on the hull after two to three days of the heading stage, and a brownish pigmentation (comparable with *bh1*) was present across the panicle (Figure 5e–h). The agronomic yield traits also revealed a significant decrease in the seed setting rate (Figure 5i).

To further verify, we constructed a complementary sequence *Os1βGlu4* along with its native promoter (1449 bp) and transformed the vector into the *bh1*. All positive transgenic lines of *bh1* recovered the white-straw hull phenotype (comparable to WT). The complementation lines did not show any brown pigmentation or lesion on the hulls (Figure 5c,d). The seed setting rate was also stored to that of the WT plants.

Agronomic yield traits of all transgenic lines were also compared at high-temperature (HT) with normal temperature (NT, Figure 5j). Consistently, the seed setting rate and the number of fertile panicles of knock-out lines were significantly decreased at a higher temperature more vigorously than at a normal temperature.

Transgenic assays have validated that the phenotype of brown pigmentation in *bh1* was due to the mutation in the *Os1βGlu4.*

Subcellular localization of *Os1βGlu4* in the cytoplasm and its expression spectrum.

The prediction of *Os1βGlu4* by targetP-2.0 server shows its subcellular localization in the cytoplasm. To confirm the subcellular localization of *Os1βGlu4*, the coding region was transferred into pCAMBIA2300-35S-eGFP vector system. The recombinant vector was transformed into rice protoplasts results, and revealed the subcellular localization of *Os1βGlu4* in the cytoplasm and nucleus (Figure 6a). This result is consistent with the previous localization of potato protoplasts [39]. The Gene expression spectrum of *Os1βGlu4* was initially retrieved from the BAR ePlant Browser (http://bar.utoronto.ca/eplant_rice/, accessed on 15 June 2022), which shows its expression during the whole growth period in rice. RT-qPCR analysis revealed the spatio-temporal expression of *Os1βGlu4,* and its highest expression was in roots at the three-leaf stage, and young panicle and leaves at maturity (Figure 6b). The GUS promoter assay was further conducted on the different tissues of transgenic plants (Figure 6c–g). These transgenic plants contained a 2 kbp promoter sequence of *Os1βGlu4* in the pCAMBIA2181-GUS reporter gene system. The GUS staining revealed the ubiquitous expression of *Os1βGlu4* in all measured tissues. However, the highest GUS expression of *Os1βGlu4* was detected in spikelets and roots. However, the GUS activity was lower in the vascular tissues of anther filaments and styles. Together, the expression spectrum revealed the ubiquitous expression of *Os1βGlu4* in all parts; however, it was preferentially expressed in young panicles and roots. 

### 2.5. Os1βGlu4 Can Hydrolyze SAG In-Vitro and InVivo

Previously, it was reported that *Os1βGlu4* could hydrolyze p-nitrohenyl (pNP)-β-d-glucoside, β-(1-3) linked oligosaccharides, salicin, and esculin in-vitro [39]. To test, the biological function of Os1βGlu4 in vivo SA was injected with different concentrations (0.1 mm, 0.5 mm, 1 mM) to the base of *bh1*’s panicle to get rid of the pigmentation (Figure 7a). The *bh1* phenotype of brown hull pigmentation *bh1* was significantly recovered to varying degrees under the exogenous treatment of SA. However, the exogenous application of ddH_2_O to *bh1* did not recover the *bh1* phenotype. 

Recovery of the mutant phenotype by the exogenous application of SA reveals that the deficiency of SA might have caused the pigmentation of hulls in *bh1*. To test this, we quantified the activity of β-Glu in WT, *bh1*, knock-out lines, and complementation lines. The activity of β-Glu was significantly reduced in *bh1* and knock-out lines (Figure 6b). As β-Glu can hydrolyze the SAG into the free form of SA. We further quantified the level of SAG and SA in *bh1* after the pigmentation (Figure 6c,d). The level of SAG was significantly higher in *bh1* than WT, and the level of SA was vice versa. Together, this data indicates that reduced β-Glu activity has produced the deficiency of SAG to SA conversion, resulting in the pigmentation of hulls in *bh1*. 

## 3. Discussion

### 3.1. Cytoplasmic Os1βGlu4 Contributes to Cellular SA Pools in Rice

In 1828, the French pharmacist Henri Leroux and Italian Chemist Raffaele Piria succeed in isolating the salicin’s crystalline form from the willow bark. The saturated water-based solution with a pH of 2.4 was named SA [42]. Until 2019, the researchers have elucidated the isochromatic and PAL pathway mechanisms for the biosynthesis of SA in plants [43]. However, the systematic recycling from an inactive SAG to active SA is regulated by cytoplasmic β-Glu, and has not been reported so far. However, the role of Os3βGlu6 and its two homologs (AtBH1 and AtBG2) in Arabidopsis was reported to increase stomatal density by increasing the content of ABA in the cellular pool [38,44,45]. OsSGT1 encodes a glucosyltransferase and catalyzes the binding of free SA to glucose to form SAG and regulates basal disease resistance [46]. Glucosyltransferase family genes were reported to play their functional role in converting SA to SAG. However, the role of β-Glu in recycling SA remained largely unknown. 

A mutant (*bh1*) from the EMS mutagenized population was identified in the current study. Forward genetic approaches were deployed to study the functional analysis of its candidate gene (Os1βGlu4). The current observation results revealed that Os1βGlu4 is essential for converting SAG to SA in the cellular pool. The loss of function of Os1βGlu4 caused the brown pigmentation in hulls of rice. 

### 3.2. SA Deficient Plants Have Brown Pigmentation on the Hulls and Showed Decreased Seed Setting Rate in Rice

The phenotypic analysis of *bh1* plants revealed the presence of brown pigmentation on the hulls of caryopsis. The presence of brown pigmentation halted the efficiency of photosynthesis by decreasing the NPR, Chl-a, and Chl-b contents in *bh1*. The *bh1* also exhibited another SA-deficient phenotype, such as sensitivity to high temperature. The brown pigmentation and low seed setting rate were restored to normal by the exogenous application of SA. These results revealed that *bh1* phenotypes were attributed to the deficiency of freely available SA in the cellular pool. A previous study revealed that plants possess a different volume of phytohormones in different parts such as the panicle [47]. It employs the additional comparative profiling of phytohormones and whether their individual or combined exogenous application can also recover the phenotype of *bh1*, as that of SA should be investigated in the future. 

The contents of MDA and H_2_O_2_ were significantly increased after the pigmentation of hulls compared to before the pigmentation. The increased accumulation of ROS causes PCD [48,49] and could be a potential source for the accumulation of plaque-like phenotype on the hulls of *bh1*. A plaque-like phenotype was developed due to the decreased removal of excessive ROS from the glume cells of *bh1*. Expression analysis of Os1βGlu4 revealed its highest expression at the tri-foil leaf stage. Previous studies have reported that SA plays a protective role against biotic and abiotic stresses. Loss of function of Os1βGlu4 results in a decrease of cellular SA, which puts the plant at more risk of biotic and abiotic stresses. These results suggest that Os1βGlu4 may also increase the resistance to abiotic and biological stresses by mediating the levels of SA and SAG at high temperatures. However, future studies would be needed to confirm this.

### 3.3. Cytoplasmic Os1βGlu4 May Affect SA Pathway and Flavonoid Pathways for Brown Hull Pigmentation

Secondary metabolites and their distinctive roles in floral organ coloration and antioxidant properties are beneficial for human health [50]. Flavonoids are mainly divided into chalcones, flavonols, anthocyanins and proanthocyanidins. Previous studies reported that methyl-SA is involved in the stimulation of phenylpropanoids and increases the biosynthesis of flavonoids [51]. Moreover, SA has also been reported in the induction of the flavonoid biosynthesis pathway in wheat [52]. Many structural genes encode the enzymes catalyzing the PAL to produce flavonoids [9]. The conserved MBW complex is thought to regulate the common pathway of flavonoid biosynthesis, especially anthocyanins and proanthocyanidins, but flavonol biosynthesis appears to be regulated only by MYB TFs [53,54]. A classic model of the C-S-A gene system has been reported in rice for hull coloration [27]. The C1 is an R2R3-MYB TF, which determines the color formation. S1 encodes a bHLH TF and controls the coloration in a tissue-specific manner, while A1 encodes the dihydroflavonol reductase, which is responsible for anthocyanin biosynthesis. In Arabidopsis, TRANSPARENT TESTA GLABRA1 (TTG1) encodes a WD40 repeat protein and is involved in anthocyanin biosynthesis and trichome development [55,56]. Previous studies have shown that AtTTG1 interacts with different R2R3-MYB and bHLH to form different MBW complexes, which play a role in anthocyanin accumulation in vegetative tissues and proanthocyanin biosynthesis during seed development. An AtTTG1 homologous gene in maize, PAC1 recovers the phenotype of the ttg1 mutant, suggesting that WD40 proteins have a conserved function between dicotyledonous and monocotyledonous plants [57]. Kala4 encodes bHLH TF and plays a tissue-specific staining function like OsC1. OsSDFR encodes dihydroflavanol reductase as OsA1 and regulates the anthocyanin biosynthesis. OsC1 and OsA1 are the key factors in color formation, which determine whether and what color can be formed in the floral organ. The loss of function of OsDFR in the blockage of cornulin and brown glumes phenotype is formed [55]. SA also plays an important role in the flavonoid pathway and regulates the response against biotic and abiotic stresses. The synthesis of flavonoids enhances the level of quercetin and catechin in tea plants in response to biological and abiotic stresses. Further investigations of flavonoid components and their detailed comparison of WT and *bh1* will be carried out in the near future.

## 4. Materials and Methods

### 4.1. Plant Material and Growth Conditions

A mutant (*bh1*) with a stable phenotype of hull pigmentation was identified from the ethyl methane-sulfonate (EMS)-mutagenized population of an indica cv. Yixiang1B. Yixiang 1B plants were used as WT plants throughout this study. The BC_1_F_2_ population was constructed by hybridization with *bh1* as a female parent and WT as a male parent for Mutmap sequencing. In addition, japonica cultivar 02428 was also used as a male parent to cross with *bh1* to construct an F_2_ for gene mapping. All experimental materials were alternately planted in the experimental fields of Sichuan Agricultural University, at Lingshui (N18.47°, E110.04°), Hainan Province, and Wenjiang (N30.67°, E104.06°), Sichuan Province.

### 4.2. Measurement of Photosynthetic Pigments 

Photosynthetic pigments were extracted from the leaves by treating them with 80% acetone at room temperature (4 °C) for 48 h according to a previous study [38]. The pigment contents were determined by spectrophotometer (Thermo, Muliskanmk3, USA) at 663 nm, 646 nm and 470 nm, respectively. Net photosynthetic rate and stomatal conductance were studied with POR plot photosynthetic apparatus (LI-6400, Lincoln LI-COR, Lincoln, NE, USA) according to a method previously described by Li et al. [58]. The photonic flux density was 1200 μmol m^−2^ S^−1^ under ambient control at a temperature of 30 °C, and the concentration of carbon dioxide was 400 ppm.

### 4.3. Determination of Antioxidants and TUNEL Assay

The quantification of antioxidants was determined by an enzyme-linked immunosorbent assay (ELISA) kit purchased from SolarBio Life Sciences Co., Ltd., Beijing, China and analyses were performed according to the instruction of the manufacturer. A hydrogen peroxide detection kit [59], malondialdehyde (MDA) assay kit [60], superoxide dismutase (SOD) activity detection kit [61] and a catalase (CAT) activity assay kit [62] were applied according to the studies reported previously. In this study, WT and mutant *bh1* plant hulls were selected to perform the TUNEL experiment before and after pigmentation according to the study of Ali et al. [48] The kits used in the experiment were purchased from Shanghai Roche Technology Co., Ltd., Shanghai, China. 

### 4.4. Genetic Analysis and Map-Based Cloning

Map-based cloning was performed with more than 700 SSR markers using 300 *bh1* individuals selected from the F_2_ population. Polymorphic bands were screened from all the chromosomes. Bulked sergeant analysis was carried out, and 256 microsatellite markers were applied for primary gene mapping according to our previous study by Liao et al. [63]. New Indel tags were developed based on the genomic differences between japonica (nipponbare) and indica (9311) genomes. A mutMap analysis was performed on 30 individuals of BC_1_F_2_ as mentioned in a previous study [41]. All primers are listed in Appendix A.

### 4.5. Gene Knock-Out, Complementation and Overexpression Analysis

A full-length sequence of the LOC_Os01g67220 genome, including its promoter (2 Kb) was cloned into the complementation vector pCAMBIA1300. The development of a single guided RNA (sgRNA) for the target sequence was done according to previous study by Ali et al. [48]. The amplified products were cloned according to the instructions mentioned in the ClonExpress™II one-step cloning kit (Vazyme, Nanjing, China). The pCAMBIA1300-LOC_Os01g67220 recombinant vector was transformed into a *bh1* mutant by Agrobacterium-mediated transformation. The genotype of positive complementation lines was tested by sequencing. Overexpression lines were developed by transforming the recombinant vector pCAMBIA1300-LOC_Os01g67220 into WT. The knock-out vector constructed in the current experiment was derived from the laboratory of Liu Yaoguang, an academician of Huazhong Agricultural University, China. The binary vector skeleton was pCAMBIA-1300, and the polyclonal site BsAI containing multiple sgRNA expression boxes was located close to the binary vector Rb. The CRISPR/Cas9 vector was transformed into the E. coli strain Trans-DH5α according to the guidelines of Ma et al. [64]. All primers are listed in Appendix A.

### 4.6. Subcellular Localization 

The targetP-2.0 server was used for subcellular localization prediction. The full-length CDS sequence of LOC_Os01g67220 was transformed into an eGFP vector having restriction sites for BamHI and Sal1. A ClonExpress™II one-step cloning kit (Vazyme, Nanjing, China) was used to clone the LOC_Os01g67220 gene into pCAMBIA2300-eGFP. The recombinant vector of pCAMBIA2300-35S-LOC_Os01g67220-eGFP was transformed into rice protoplast according to a method described by Zhang et al. [65]. The protoplasts were incubated with 200 nM Mitotracker Red CMXROS (Invitrogen, Carlsbad, CA, USA) for 15 min at 37 °C. The laser-scanning confocal microscope (Nikon A1, Nikon, Japan) was used to detect the fluorescence signals of rice protoplasts. 

### 4.7. GUS Activity Staining Assay 

β-glucuronidase (GUS) activity was tested using a GUS staining kit purchased from Beijing Solaibao Biotechnology Co., Ltd., Beijing, China, X-GluC staining solution was prepared by dissolving the X-GluC powder in a GUS buffer and stored at 4 °C. Plant tissues were collected and immersed in the staining solution and incubated at 37 °C for 8 h. After incubation, the tissue was treated with 75% ethanol. The clean tissue samples were viewed and photographed under an anatomical microscope (Leica S8APO, Wetzlar Germany).

### 4.8. Determination of β-Glucosidase Activity

β-glucosidases decompose p-nitrobenzene-β-D-glucopyranoside to p-nitrophenol, with the maximum absorption peak at 400 nm. The activity of β-glucosidases was calculated by measuring the absorption value in a spectrophotometer. Enzyme solutions were extracted from hulls according to the instructions given in the Micro-β-Glucosidase assay kit according to Chen et al. [66] 0.2 g of tissue was added to 1 mL extract followed by ice bath homogenization. The mixture was centrifuged (15,000× *g*) at 4 °C for 20 min, and the supernatant was taken and placed on ice to be measured. The standard curve was set up and β-glucosidases activity was calculated according to the formula below.

β-glucosidases activity (U/g_fresh weight_) = (Y × V_inverse total_) ÷ (W × V_sample_ ÷ V_sample total_) ÷ T = 20 × Y ÷ W. The enzyme activity per unit is defined as 1 nmol of p-nitrophenol produced per gram of tissue per hour.

### 4.9. In-Vitro/In-Vivo SAG to SA Hydrolysis

The total protein of the fresh panicle was extracted by Plant Total Protein Extraction Kit (Beijing Kulaibo Technology Co., Ltd. Beijing, China) and used for β-glucosidase activity determination. Fresh panicles (0.1 g) were ground into powder in liquid nitrogen, added with 1 mL total plant protein extract and transferred to a 1.5 mL centrifuge, followed by centrifugation at 4 °C 12,000 rpm for 10 min. A crude enzyme solution (0.4 mL enzyme solution, 8 µL 2 nM SAG and 1.2 mL 0.1 M citrate buffer) was purchased from OlChemlm, Olomouc, Czech Republic. After incubation at 30 °C for 30 min, the solution was adjusted to a pH of 3 and extracted with ethyl-acetate. A methanol solution was prepared and configured as the final concentration of 0.1 ng/mL, 0.2 ng/mL, 0.5 ng/mL, 2 ng/mL, 5 ng/mL, 20 ng/mL, and 50 ng/mL standard curve and each point contained 10 ng/mL internal standard curving solution. The organic phase was vaporized using HPLC-MS/MS series (AB, SCIEX-QTRP6500, Framingham, MA, USA) and analyzed on a Poroshell 120 SB-C18 reverse chromatographic column. Elutions were performed using both Solvent A (methanol/0.1% formic acid) and solvent B (water/0.1% formic acid, 0–1 min 20%, 1–3 min 20–50%, 3–9 min 50–80%, 9–10.5 min 80%, 10.5–10.6 min 80–20%, 10.6–13.5 min 20%) at a flow rate of 0.3 mL min^−1^. At the same time, ESI positive and negative ion modes were used for MRM detection. The gas screen pressure was 15 psi, the spray voltage was ±4500 V, and the atomizing gas pressure was 65 psi. The monitoring of SAG and SA enzymes at 400 nm and the activity unit (U/mol) was measured at 0.01 mol of SAG per hour.

### 4.10. Quantitative Real-Time PCR Analysis

RT-qPCR was performed on a CFX96™ Real-Time System, Bio-RAD (Hercules, CA, USA) using HiScript^®^ ⅡQ RT SuperMix (Vazyme Technology Company, Nanjing, China). The reaction system of RT-qPCR was as follows: AceQ^®^ qPCR SYBR^®^ Green Master Mix 10 μL, 10 μM Primer 0.8 μL, ROX Reference Dye 0.4 μL, cDNA 2.0 μL, RNase-free Water 6.8 μL in a total volume 20 μL. The sample and the reference gene ACTIN were repeated thrice, and No Template Control was repeated twice. A quantitative expression analysis was carried out using the Bio-Rad CFX Manager V2.0 software of quantitative PCR instrument, and the algorithm was 2^−^^△△CT^ method.

## Figures and Tables

**Figure 1 ijms-23-10646-f001:**
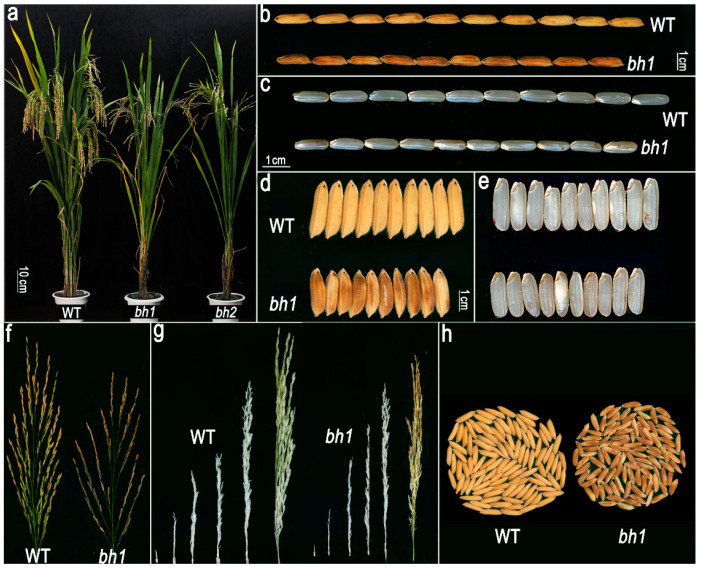
Phenotype observation and agronomic traits of *brown hull1 (bh1)*. (**a**) Plant morphology of a Wild Type (WT) plant showing normal white-straw (left) while *bh1* and *bh2* plant (right) showed pigmentation on hull, (**b**–**e**) Mature grains and caryopsis of WT and *bh1*, (**f**) expanded view panicle of WT and *bh1*, (**g**) Different stages of a panicle of WT and *bh1*, (**h**) Morphology and appearance of bulk grains of WT and *bh1*.

**Figure 2 ijms-23-10646-f002:**
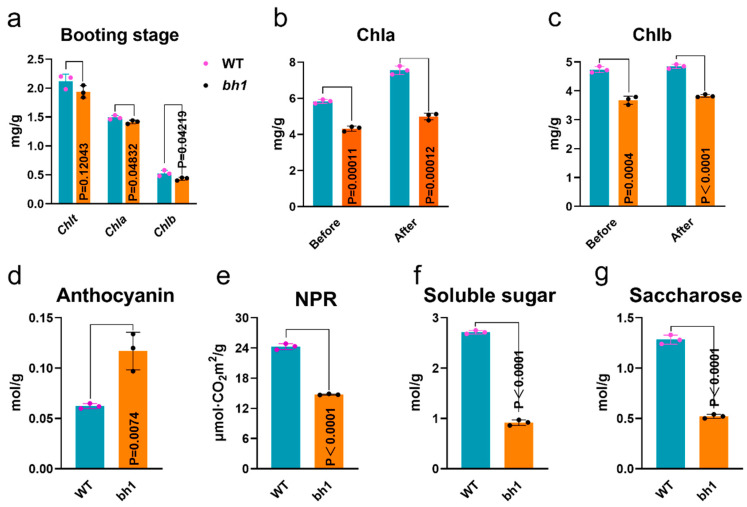
Comparison of photosynthetic parameters, anthocyanins and sugars contents of WT (wild type) and *brown hull 1 (bh1).* (**a**) Determination of photosynthetic pigment at booting stage (**b**) chlorophyll a, (**c**) chlorophyll b, (**d**) anthocyanin content in the hull, (**e**) net photosynthetic rate of mature leaves, (**f**) contents of soluble sugar (**g**) saccharose content in the hull.

**Figure 3 ijms-23-10646-f003:**
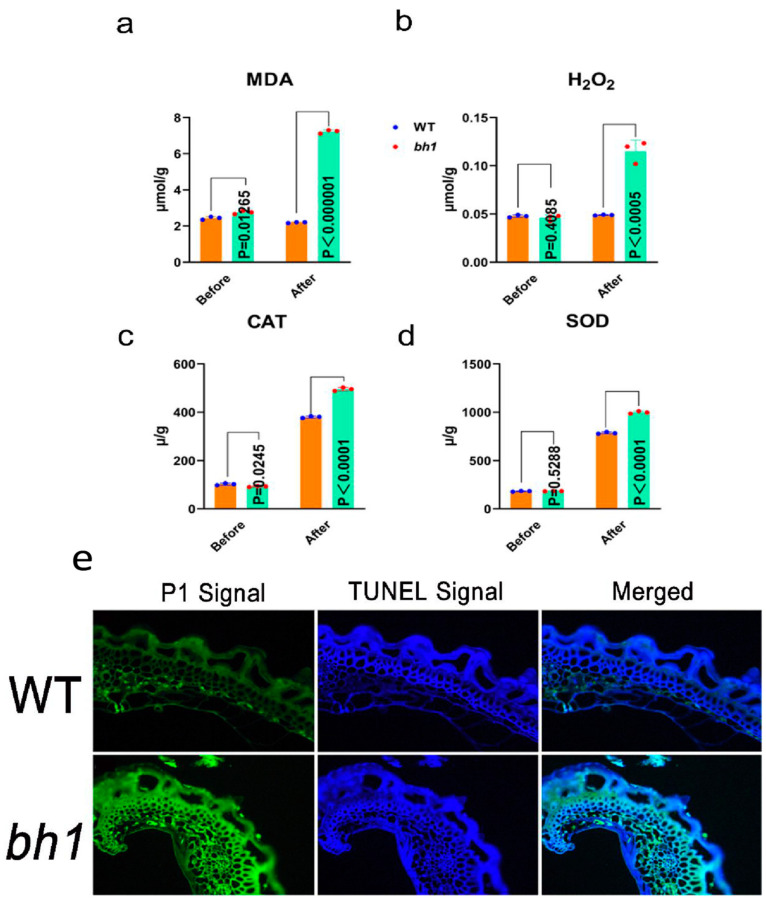
Comparison of reactive oxygen species (ROS) in the hulls of *brown hull1 (bh1)* and Wild type (WT). (**a**) Determination of malondialdehyde, (**b**) hydrogen peroxide, (**c**) catalase (**d**) superoxide dismutase contents in hulls of WT and *bh1* (**e**) TUNEL assay showing DNA fragmentation in hulls of WT and *bh1*. Propidium iodide (PI) produces green fluorescence in the tissues of hulls, while a merged view (blue and green fluorescence) displays the positive signals of TUNEL.

**Figure 4 ijms-23-10646-f004:**
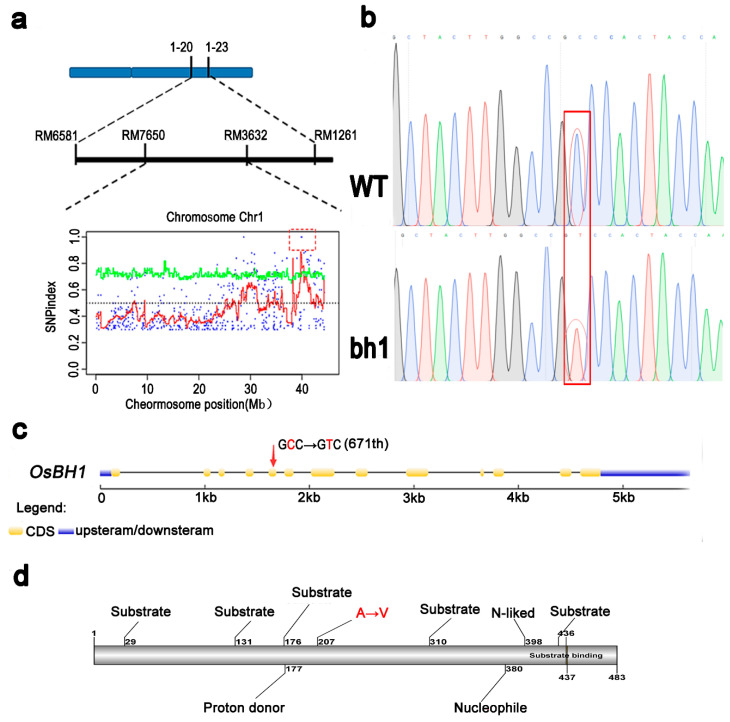
Primary mapping and the Mutmap assay of *(brown hull1) bh1.* (**a**) SSR marker and MutMap assays, (**b**) Chromatogram showing the position of SNP in *bh1* (**c**) Gene structure analysis, (**d**) protein structure analysis, where (**c**) the SNP and (**d**) the amino acid site are highlighted in red.

**Figure 5 ijms-23-10646-f005:**
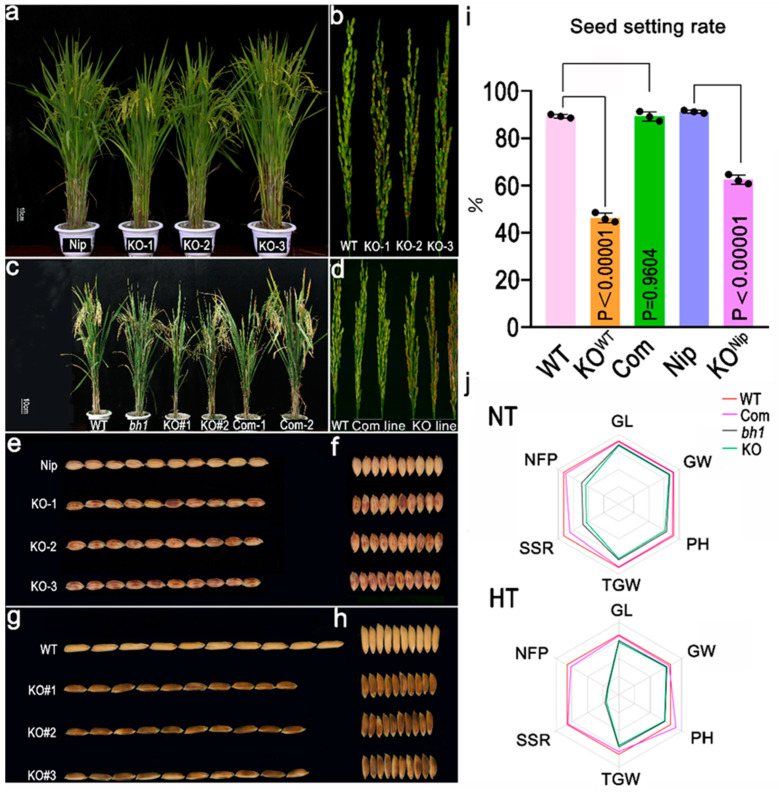
Transgenic assays of complementation and knockout to validate the function of Β-glucosidase (Os1βGlu4). (**a**) Plant morphology, (**b**) panicle morphology of Nipponbare (Nip), KO^Nip^-1, KO-2^Nip^, KO-3^Nip^ (**c**) plant morphology of (**d**) panicle morphology of WT, *bh1*, KO^WT^#1, KO^WT^#2, complementation line-1, complementation line-2 (left to right, respectively), (**e**) image showing grain length, (**f**) grain width of Nip, KO^Nip^-1, KO-2^Nip^, KO-3^Nip^ (from top to bottom), (**g**) image showing grain length, (**h**) grain width of WT, KO^WT^#1, KO^WT^#2, KO^WT^#3 (from top to bottom), (**i**) Graph showing the comparison of seed setting rate of WT, KO^WT^, Complementation, Nip, KO^Nip^, (**j**) Comparison of agronomic traits of WT, Complementation, *bh1*, KO line, where NT: Normal temperature (top), HT: High temperature (bottom), GL: Grain length, GW: Grain width, PH: Plant height, TGW: Thousand-grain weight, SSR: Seed setting rate, NFP: Number of filled grains per panicle.

**Figure 6 ijms-23-10646-f006:**
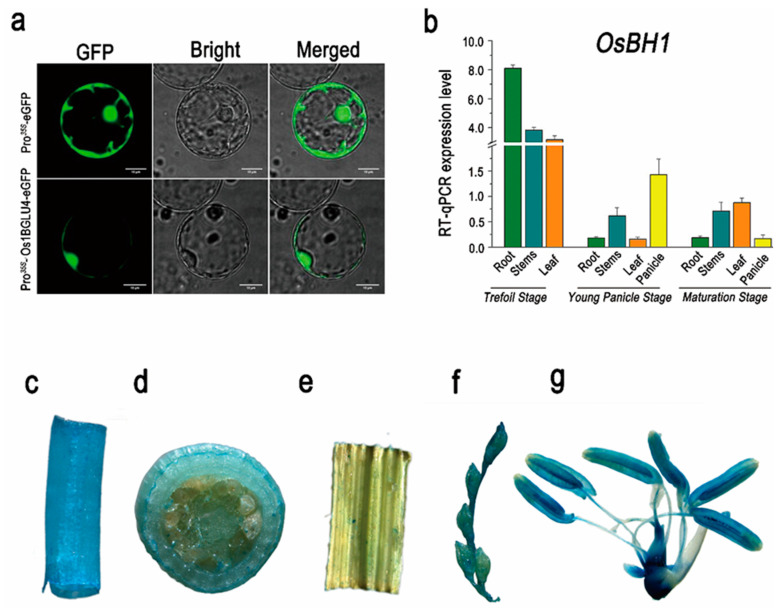
**Subcellular localization, relative expression and GUS promoter assay of *β-glucosidase (Os1βGlu4)***. (**a**) Subcellular localization of *Os1βGlu4*-eGFP fusion protein, (**b**) Relative expression level of *Os1βGlu4* in different tissues at trefoil leaf stage, young panicle stage and maturation stage, GUS promoter expression analysis in (**c**) root, (**d**) stem, (**e**) leaf, (**f**) young spikelets and (**g**) floret.

**Figure 7 ijms-23-10646-f007:**
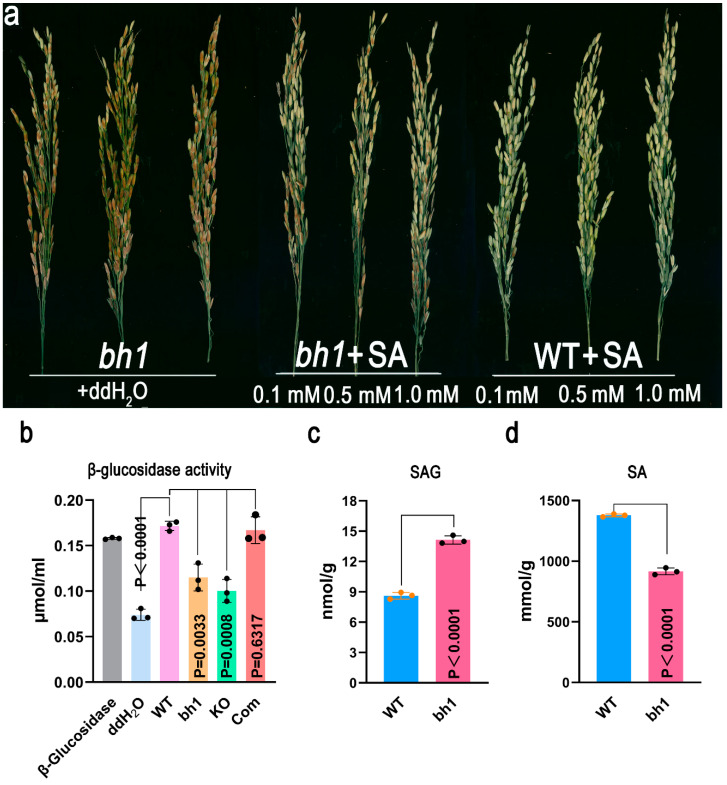
The exogenous application of SA can rescue the *(brown hull1) bh1* phenotype and in vivo hydrolysis activity of β-glucosidase (Os1βGlu4). (**a**) Phenotype of panicles after the exogenous application of SA to *bh1* and WT, with ddH_2_O used as a control, (**b**) comparison of β-glucosidase activity in WT, *bh1,* knockout line and complementation lines, (**c**) The content of SAG, and (**d**) SA in WT and *bh1*.

## Data Availability

The datasets supporting conclusions have been provided as Appendix A.

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
