# Peer review of "Rice β-Glucosidase 4 (Os1βGlu4) Regulates the Hull Pigmentation via Accumulation of Salicylic Acid"

_ijms, 2022, doi:10.3390/ijms231810646_

Round 1

Reviewer 1 Report

.    The authors found an EMS-generate mutant bh1 showed decreased contents of SA in hulls, lower photosynthesis rate, and high-temperature sensitivity. Moreover, a brown pigmentation present on the hulls caused a significant decrease in seed setting rate. Genetic analysis revealed the mutation gene is Os1βGlu4. The knock-out lines display the similar phenotype with bh1, and overexpression and complementation lines of Os1βGlu4 restore the phenotype of bh1. In contrast to WT, bh1 could not hydrolyze SAG into SA in-vivo. The results revealed a novel role of Os1βGlu4 the accumulation of flavonoids in hulls by regulating the level of free SA in the cellular pool.

.          This is a more meaningful study, and the findings are more systematic and complete,but some data should be provided or improved.

1.     The figure notes from 2 to 7 are same to figure notes 1, the authors showed correct.

2.     Figure 3 have two d in the picture.

Author Response

The authors found an EMS-generate mutant bh1 showed decreased contents of SA in hulls, lower photosynthesis rate, and high-temperature sensitivity. Moreover, a brown pigmentation present on the hulls caused a significant decrease in seed setting rate. Genetic analysis revealed the mutation gene is Os1βGlu4. The knock-out lines display the similar phenotype with bh1, and overexpression and complementation lines of Os1βGlu4 restore the phenotype of bh1. In contrast to WT, bh1 could not hydrolyze SAG into SA in-vivo. The results revealed a novel role of Os1βGlu4 the accumulation of flavonoids in hulls by regulating the level of free SA in the cellular pool.

This is a more meaningful study, and the findings are more systematic and complete,but some data should be provided or improved.

We oblige the anonymous reviewer for his time and effort, as the revised version of the manuscript has been substantially improved with your suggestions. Our point-to-point responses to the reviewer’s comments are given below.

To improve the language and Grammar of the manuscript, we have carefully proofread and revised the manuscript with the help of all authors. The sentence structure has been thoroughly checked and improved in the revised version. Changes have been highlighted in red in the revised version.

  1. The figure notes from 2 to 7 are same to figure notes 1, the authors showed correct. Response: Sorry for the major mistake, we have revised all the figure legends according to figures.

  1. Figure 3 have two d in the picture.

Response: Sorry for mistake, we have corrected this mistake in the revised version.

Reviewer 2 Report

Manuscript Number ijms-1898988, titled:

Β-glucosidase (Os1βGlu4) regulates the hull pigmentation via accumulation of Salicylic acid.

Review 1 – 23 Aug 2022

Dear Editor of International Journal of Molecular Sciences

 This study investigated on Β-glucosidase (Os1βGlu4) regulates the hull pigmentation via accumulation of Salicylic acid. The manuscript has to be improved. The introduction section has to be better presented and M&M section has to be better decribed. The manuscript has to be presented as per of International Journal of Molecular Sciences guidelines. Inaccuracies in the manuscript.

I suggest a major revision.

To the Authors (in detail):

  1. Please apply the type of the paper of of International Journal of Molecular Sciences, download it and carefully apply it;
  2. Introduction section, page 2, from line 50, when you discuss about flavonoids, indicate the plant containing flavonoids and anthocyanins and their content. Support this statement with proper and recent references. Please, find, read and discuss, at least: [X1, X2, X3]:

X1. HPLC-DAD detection of changes in phenol content of red berry skins during grape ripening.

European Food Research and Technology, 237 (4) 555-564 (2013)

DOI: 10.1007/s00217-013-2033-7

X2. Physico-chemical Stability of Blood Orange Juice during Frozen Storage.

International Journal of Food Properties 20:sup2, 1930-1943 (2017). https://doi.org/10.1080/10942912.2017.1359184

X3. Comparison of Flavonoid Composition of Red Raspberries (Rubus idaeus L.) Grown in the Southern United States.

J. Agric. Food Chem. 2012, 60, 23, 5779–5786

https://doi.org/10.1021/jf203474e

  1. Figure 2f: Soluble suger?
  2. Sub-section 4.6, line 434 and in the whole manuscript, tables and figures, when you indicate a temperature, separate the numeric value from the symbol: 370 °C and not 37°C;
  3. Sub-section 3.9, line 459 and in the whole manuscript, tables and figures:  separate the numeric value from the unit: 0.1 g and not 0.1g;
  4. Sub-section 3.9, line 461 and in the whole manuscript, sometime you have written ml and sometime mL. Please, be consistent. I suggest mL, in addition, separate: 0.4 mL;
  5. Line 461, you have used a wrong symbol for microliters. The correct letter from the old Greek alphabet is μ, for μL;
  6. Line 461 and in the whole manuscript, tables and figures, 10 min and not 10min;
  7. Line 463, delete one space before …. the solution;
  8. Line 472, 15 psi, separate;
  9. Line 476, delete one space before.. Bio-rad;
  10. Line 479 and in the whole manuscript, 0.8 μL, separate;
  11. The references are not exactly current;
  12. Please, write in blue color or evidence differently the corrections you will do

Regards

Author Response

Β-glucosidase (Os1βGlu4) regulates the hull pigmentation via accumulation of Salicylic acid.

Review 1 – 23 Aug 2022

Dear Editor of International Journal of Molecular Sciences

 This study investigated on Β-glucosidase (Os1βGlu4) regulates the hull pigmentation via accumulation of Salicylic acid. The manuscript has to be improved. The introduction section has to be better presented and M&M section has to be better decribed. The manuscript has to be presented as per of International Journal of Molecular Sciences guidelines. Inaccuracies in the manuscript.

We oblige the anonymous reviewer for his time and effort, as the revised version of the manuscript has been substantially improved with your suggestions. Our point-to-point responses to the reviewer’s comments are given below.

To improve the language and Grammar of the manuscript, we have carefully proofread and revised the manuscript with the help of all authors. The sentence structure has been thoroughly checked and improved in the revised version. Changes have been highlighted in red in the revised version.

I suggest a major revision.

To the Authors (in detail):

Please apply the type of the paper of of International Journal of Molecular Sciences, download it and carefully apply it;

 Response: We have applied the formatting in the text. However, according to the handling editor, the monogram and numbering of the left side would be applied after the acceptance.

Introduction section, page 2, from line 50, when you discuss about flavonoids, indicate the plant containing flavonoids and anthocyanins and their content. Support this statement with proper and recent references. Please, find, read and discuss, at least: [X1, X2, X3]:

X1. HPLC-DAD detection of changes in phenol content of red berry skins during grape ripening.

European Food Research and Technology, 237 (4) 555-564 (2013)

DOI: 10.1007/s00-7

X2. Physico-chemical Stability of Blood Orange Juice during Frozen Storage.

International Journal of Food Properties 20:sup2, 1930-1943 (2017). https://doi.org/10.1080/10942912.2017.1359184

X3. Comparison of Flavonoid Composition of Red Raspberries (Rubus idaeus L.) Grown in the Southern United States.

  1. Agric. Food Chem. 2012, 60, 23, 5779–5786

https://doi.org/10.1021/jf203474e

 Response: Thanks for the comment, we have carefully read the mentioned articles and found these studies were not aimed to explore the molecular mechanism of flavonoids but only physiochemical properties under treatments were studied. So, we are sorry and have not cited two of them. But for your concern, we have cited relevant and recent studies.

Figure 2f: Soluble suger?

Response: Thanks for pointing mistake, we have corrected this mistake.

Sub-section 4.6, line 434 and in the whole manuscript, tables and figures, when you indicate a temperature, separate the numeric value from the symbol: 370 °C and not 37°C;

Response: We have placed a space between all integers and their measuring units.

Sub-section 3.9, line 459 and in the whole manuscript, tables and figures:  separate the numeric value from the unit: 0.1 g and not 0.1g;

Response: We have carefully revised the whole manuscript and separated integers and measuring units by space.

Sub-section 3.9, line 461 and in the whole manuscript, sometime you have written ml and sometime mL. Please, be consistent. I suggest mL, in addition, separate: 0.4 mL;

Response: We have carefully revised the whole manuscript and separated integers and measuring units by space.

Line 461, you have used a wrong symbol for microliters. The correct letter from the old Greek alphabet is μ, for μL;

Response: Sorry for the mistake we have replaced u with μ.

Line 461 and in the whole manuscript, tables and figures, 10 min and not 10min;

Response: We have carefully revised the whole manuscript and separated integers and measuring units by space.

Line 463, delete one space before …. the solution;

Response: Deleted

Line 472, 15 psi, separate;

Response: We have carefully revised the whole manuscript and separated integers and measuring units by space.

Line 476, delete one space before.. Bio-rad;

Response: Deleted

Line 479 and in the whole manuscript, 0.8 μL, separate;

Response: We are sorry for the mistake, we have carefully revised the whole manuscript and separated integers and measuring units by space.

The references are not exactly current;

Response: We have cited the recent and relevant articles in revised version.

Please, write in blue color or evidence differently the corrections you will do

Response: Thanks for pointing out mistakes, changes are highlighted in red text in revised version.

Round 2

Reviewer 2 Report

Manuscript Number ijms-1898988, titled:

 Manuscript Number ijms-1898988, titled:

 Rice β-glucosidase (Os1βGlu4) regulates the hull pigmentation via accumulation of Salicylic acid.

Review 2 – 2 Sept 2022

Dear Editor of International Journal of Molecular Sciences

 This study investigated on Rice β-glucosidase (Os1βGlu4) regulates the hull pigmentation via accumulation of Salicylic acid. The Authors have included many of my comments.

However:

  1. the Authors have to use the Microsoft Word Template: Google> international Journal of Molecular Sciences template;
  2.  the references section is not organized as per IJMS guidelines: 1.the journal name has to be abbreviated (digit google > journal name abbreviations);
  3. the first letter of each word of the journal name has to be written in capital letter;
  4. Refs 6, 45 and in the whole section and manuscript: the scientific name has to be italicized;
  5. ref 29 has to be completed;
  6. Ref 40 has to be completed;
  7. Ref 48 has to be completed;

Please, write in red color or evidence differently the corrections you will do

I suggest a minor revision.

Regards

Author Response

Please find the response letter in the attachment.
